



# Analysis of 2D airglow imager data with respect to dynamics using machine learning

René Sedlak[1], Andreas Welscher[1,2], Patrick Hannawald[2], Sabine Wüst[2], Rainer Lienhart[3], and Michael Bittner[1,2]

[1] Institute of Physics, University of Augsburg, Augsburg, Germany
[2] German Remote Sensing Data Center, German Aerospace Center, Oberpfaffenhofen, Germany
[3] Institute of Computer Science, University of Augsburg, Augsburg, Gemany

*Correspondence to*: Sabine Wüst (sabine.wuest@dlr.de)

**Abstract.** We demonstrate how machine learning can be easily applied to support the analysis of large amounts of OH* airglow

imager data. We use a TCN (temporal convolutional network) classification algorithm to automatically pre-sort images into the three categories "dynamic" (images where small-scale motions like turbulence are likely to be found), "calm" (clear-sky images with weak airglow variations) and "cloudy" (cloudy images where no airglow analyses can be performed). The proposed approach is demonstrated using image data of FAIM 3 (Fast Airglow IMager), acquired at Oberpfaffenhofen, Germany between 11 June 2019 and 25 February 2020, achieving a mean average precision of 0.82 in image classification.

The attached video sequence demonstrates the classification abilities of the learned TCN.

Within the "dynamic" category, we find a subset of 13 episodes of image series showing turbulence. As FAIM 3 exhibits a high spatial (23 m pixel$^{-1}$) and temporal (2.8 s per image) resolution, turbulence parameters can be derived to estimate the energy diffusion rate. Similar to the results the authors found for another FAIM station (Sedlak et al., 2021), the values of energy dissipation rate range from 0.03 to 3.18 W kg$^{-1}$.

**Keywords**

*Gravity waves, turbulence, airglow imager, big data, machine learning, neural networks, temporal convolutional networks*

## 1 Introduction

Airglow imagers are a well-established method for studying UMLT (upper mesosphere – lower thermosphere) dynamics. As

the short-wave infrared (SWIR) radiation of excited hydroxyl (OH*) between approx. 82 and 90 km height (von Savigny, 2015; Wüst et al., 2017b, 2020) are known to be the brightest diffuse emissions during night-time (Leinert et al., 1998; Rousselot et al., 1999), atmospheric dynamics is observed using airborne (Wüst et al., 2019) or ground-based SWIR cameras (Taylor, 1997; Nakamura et al., 1999; Hecht et al., 2014; Pautet et al., 2014; Hannawald, 2016, 2019; Sedlak et al., 2016, 2021). OH* measurements are also possible from satellite (see table 1 of Wüst et al. (2023) for limb instruments). They can

be made in limb or nadir. The limb measurements address indeed mainly the SWIR range and are mostly used for deriving


information about the OH* layer height and thickness. Nadir-looking instruments, however, such as VIRRS DNB (Day/Night Band nightglow imagery from the Visible/Infrared Imaging Radiometer Suite) on board Suomi NPP (Suomi National Polar orbiting Partnership) and JPSS-1 (Joint Polar Satellite System-1) which have been used for analyses of atmospheric dynamics until now are measuring in the VIS (visible) range. In contrast to imager systems using an all-sky lens, which enable us to

observe the entire dynamical situation of the nocturnal sky, operating an imager with a lens of long focal length and narrow aperture angles provides the opportunity to observe small-scale dynamical features in the UMLT with a high spatial resolution. This includes instability features of gravity waves, such as 'ripples' (Peterson, 1979; Taylor and Hapgood, 1990; Li et al., 2017), but also turbulence (Hecht et al., 2021; Sedlak et al., 2016, 2021). Former studies at Oberpfaffenhofen (Sedlak et al., 2016) and Otlica, Slovenia (Sedlak et al., 2021) using the high-resolution airglow imager FAIM 3 have shown that the

observation of turbulent episodes in the OH* layer is possible with this kind of instrument.

Turbulence marks the end of the life cycle of breaking gravity waves (Hocking, 1985). Having become dynamically or convectively unstable, the wave can no longer exist and eventually breaks down developing eddies. Within this inertial subrange of turbulence energy is cascaded to smaller and smaller structures until it is dissipated via viscous damping, causing a heating effect on the atmosphere. Observing turbulence episodes with high-resolution airglow imagers, the respective energy

dissipation rate can be derived from the image series by reading out the typical length scale $L$ of the eddies and the root-mean-square velocity $U$, i.e., the velocity of single eddy patches relative to the background motion (Hecht et al., 2021; Sedlak et al., 2021).

The energy dissipation rate $\epsilon$ is given by

$$\epsilon = C_\epsilon \frac{U^3}{L} \quad (1)$$

(Chau et al., 2020), where $C_\epsilon \approx 1$ (Gargett, 1999). The results in Sedlak et al. (2021) suggest that the heating rate driven by the turbulent breakdown of gravity waves can be locally and within a few minutes as large as the daily chemical heating rates in the mesopause region (Marsh, 2011). Thus, one has to assume, that this dynamically driven effect is of great importance for the energy budget of the atmosphere and needs to be included realistically into modern climate models.

In order to derive statistically resilient and also global information about gravity wave energy deposition in the UMLT, more

and more high-resolution airglow imagers need to be deployed at different locations around the world. The largest challenge is to identify turbulence episodes in a rapidly growing data set of airglow images. While in former studies turbulence episodes were found by manual inspection (Sedlak et al., 2016, 2021), this will not be feasible anymore with much larger amounts of data.

When it comes to image recognition, artificial intelligence (AI) is a field that has seen tremendous progress in recent years.

(Fujiyoshi et al., 2019; Horak & Sablatnig, 2019; Guo et al., 2022). In particular, algorithms using Neural Networks (NN) show a very good performance in identifying different objects in images, and also have a quite efficient computation time. Using these methods to detect turbulence in airglow images present several challenges that complicate the use of off-the-shelf image recognition algorithm:



- Turbulent movement manifests itself in a wide variety of shapes and structures
- Structures in the OH* layer appear blurred and contrasts are strongly dominated by clouds
- Turbulence can often only be identified in the dynamic course of a video sequence; single images of a turbulent episode are easily confused with clouds
- The number of images showing turbulence is much smaller than the number of images showing no turbulence, thus there is an essential disbalance of available training data for the different categories.

All these aspects have the consequence that a direct extraction of turbulent episodes from the entire measurement data set is hardly possible. However, some existing approaches exhibit promising advantages that could help finding turbulence episodes. In this work, we show how NN-based methods can be combined into an algorithm that is easy to use and performs well in strongly reducing the data basis where turbulence can likely be found. We demonstrate the application and performance of this practical approach on OH* airglow image data of the FAIM 3 (Fast Airglow Imager) instrument acquired at Oberpfaffenhofen,
Germany between 11 June 2019 and 25 February 2020.

Our goal is to explain our approach in a way that it is relatively easy to apply for airglow scientists who are not specialized in AI. Therefore, the description of the classification algorithm (section 3) is in more detail than for example the introduction of the airglow instrument or the data preparation (section 2).

**2 Instrumentation and data preparation**

The OH*-airglow imager FAIM 3 (Fast Airglow Imager) is based on the short-wave infrared (SWIR) camera CHEETAH CL by Xenicx nv. The system is already described in Sedlak et al. (2016), therefore, only the most important information is given here.

The SWIR camera consists of a 512 x 640 pixels InGaAs focal plane array, which is sensitive to infrared radiation with a wavelength between 0.9 and 1.7 µm. Images are acquired automatically during each night (solar zenith angle > 100°) with a
temporal resolution of 2.8 s. Since June 2019 measurements have been performed at the DLR site at Oberpfaffenhofen (48.09°N, 11.28°E), Germany with a zenith angle of 34° and an azimuthal angle of 204° (SSW direction). Due to the aperture angles of 5.9° and 7.3° this results in a trapezium-shaped field of view (FOV) in the mean OH* emission height at ca. 87 km with size 175 km² (13.0 – 13.9 km x 13.1 km) and a mean spatial resolution of 23 m·pixels$^{-1}$. The field-of-view (FOV) is located ca. 80 km south of Augsburg, Germany.

The data basis used in this work consists of nocturnal image series acquired between June, 11th 2019 and February, 25th 2020. During this period, measurement have been performed in 258 nights. Analyzing keograms, 95 (37 %) of these nights show complete cloud coverage and do not allow analyses based on airglow observations. The remaining 163 nights exhibit a clear sky either all the time or at least temporarily (at least ca. 30 min), so that the OH* layer is visible in a total of 188 episodes (it is possible that one night has more than one clear episode as soon as it is interrupted by cloudy episodes).





The images are prepared for analysis by performing a flat-field correction and transforming each of them onto an equidistant grid (for further details see Hannawald et al., 2016). In order to completely remove any pattern remnants, such as reflections of the objective lens in the window, the average image, a pixel-wise mean of all images, of each episode is subtracted from the individual images.

## 3 Image classification with Neural Networks

### 3.1 Label classes

When looking at the temporal course of the image data, three main types of observation can be distinguished, which we use as label classes for the classification algorithm (typical examples of these label classes are shown in Figure 1):

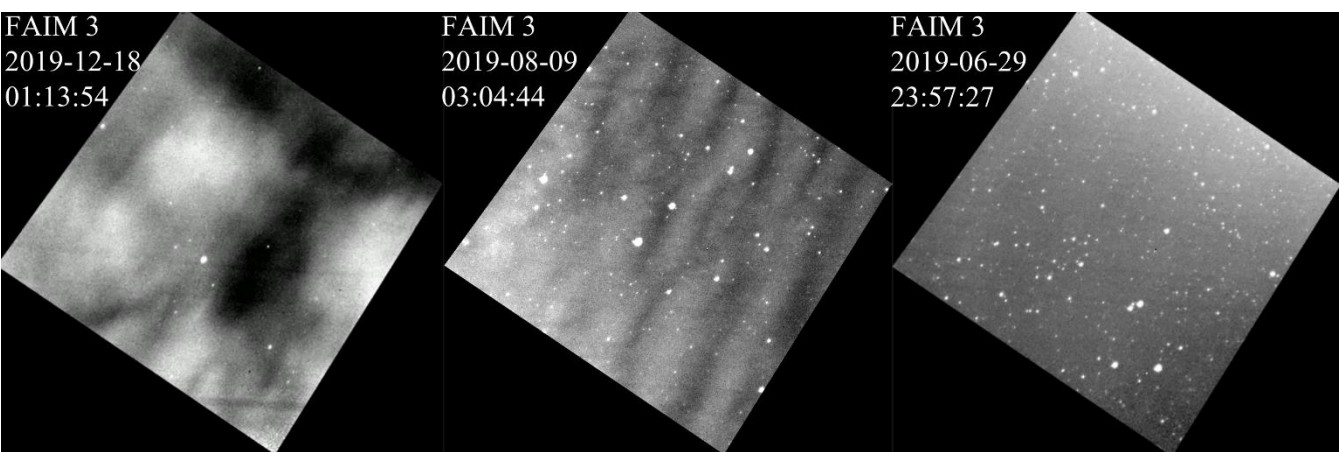

**Figure 1: Typical Examples of the three label classes: a) "cloudy", b) "dynamic" and c) "calm".**

• "Cloudy". Episodes of clouds or cloud fragments moving through the image (Figure 1a). These cloudy episodes are too short or too faint to be recognized during keogram analysis (see Section 2). The image series are characterized by sharp contrasts and fast movement of coherent structures. Often (but not necessarily) stars are covered by the clouds.

• "Dynamic". Cloud-free episodes with pronounced moving OH*-airglow structures (Figure 1b), including waves and eddies. OH* dynamics can be distinguished from cloud movement due to slower velocities, blurrier edges and (except

for extended wave fields) more isotropic movements.

• "Calm". Cloud-free episodes with weak movement in the OH* layer (Figure 1c). The images appear quite homogeneous and hardly change in the temporal course.

The goal of the classification algorithm is to automatically and reliably identify the "dynamic" label class. In a subsequent step, which is not part of this work, these episodes can then be analyzed with respect to turbulence.



## 3.2 Image features

We calculated a set of one-dimensional features for each image series that we believe help distinguishing the categories introduced in section 3.1.

The mean value and standard deviation are calculated for every image. The mean value feature was calculated based on the assumption that "cloudy" episodes have a higher intensity than "calm" episodes due to reflections from ground lights or the moon. The standard deviation of the label class "calm" is expected to be lower, whereas the label class "clouds" is expected to have higher values, since clouds show very intense cloud structures and OH* airglow outside of cloud structures, which is less intense. Mean and standard deviation of the label class "dynamic" are to be expected between "calm" and "dynamic". We refer to these two features (mean value and standard deviation) as the basic features.

They are supplemented by three texture-based features derived from the grey level co-occurrence matrix (GLCM) as described in Zubair and Alo (2019). Homogeneity is the first of these three texture-based features and a way of measuring the similarity of neighboring pixels in an image. If its value is particularly high, it suggests a high similarity of adjacent pixels (Zhou et al. 2017). This may indicate a positive correlation with the label class "calm". The second texture-based feature, dissimilarity, is inversely correlated to homogeneity and thus could help to identify the episodes containing a lot of motion. The third texture-based feature, uniformity, is particularly high, if the image has uniform structures. On the other hand, this value is very small as soon as the image contains heterogeneous structures.

Additionally, features which are based on a 2-dimensional Fast Fourier Transform (2d-FFT) as described in Hannawald et al. (2019) are derived for each image. They are called "psd" feature group in the following. As in Sedlak et al. (2021), the 2d-FFT is applied to a squared cut-out centered at the image center with side length 406 pixels (9.3 km). This results in 2-dimensional spectra, which depend on the zonal and the meridional wave number. They are integrated over these wave numbers such that the power spectral density (PSD) in dependence of the horizontal wave number $k$ is derived. According to Kolmogorov (1941), the log(PSD)-log(k) shows different slopes, which depend on whether the observed field is in the buoyancy (dominating energy transport by waves), the inertial (energy cascades to smaller scales) or the viscous subrange (viscous damping of movements). Therefore, the feature slope is derived as the linear fit in the log(PSD)-log($k$) plot. Then, the PSD is integrated over all $k$ and the change of this value per timestep is calculated. This feature is called DiffIPSD (differences of integrated PSD) and takes the fact into account that, for example, clouds are causing stronger fluctuations over time than during clear sky episodes. Last, the PSD is integrated over all $k$ and over the whole night. This feature is denoted IPSD (integrated PSD).

In total, we calculate eight different features for each image.



### 3.3 Data basis for the classification algorithm

The features introduced in the last subsection were calculated for every fifth image. This results in a temporal resolution of 14 seconds and data set of approximately 240,000 time steps. About 105,000 time steps are assigned to the label class "calm", 65,000 to "cloudy" and 70,000 to "dynamic". This data set is divided into three parts: training, validation and test data. The partition is performed as follows: first, the list of all measured nights is arranged chronologically and divided into parts with ten measured nights each. From these parts, one measured night is randomly selected and assigned to the test data. From the remaining nine measured nights, two are randomly assigned to the validation data. The remaining seven measured nights are assigned to the training data. This results in approximately 70% training data, 20% validation data and 10% test data by looking at the total number of time steps. All features of the three datasets were independently normalized to the range 0 to 1. Before normalization, the outliers (lowest and highest 0.05 quantile) were replaced by the highest value of the lowest 0.05 quantile and the lowest value of the highest 0.05 quantile.

The training data set is used to train a neural network, and at the end of an epoch (one training of the whole training data set) the result and the learning progress are checked using the validation data set. After running through all 100 epochs and additional possible manual adjustments to the classification procedure, the final quality of a classifier is determined on the test data set. This procedure serves, among other things, to avoid overfitting to the training data set. In order to use this procedure properly, the training, validation and test data must be different from each other. In our case, this is ensured by dividing the complete data set into training, validation and test data set by complete nights and not by individual parts of a night. Features from one night may be more similar to each other than features from different nights which could lead to unnoticed overfitting if parts of a night were used for the partitioning into the training, validation and test data set.

### 3.4 Classification algorithm

A neural network consists of an input and an output layer as well as one or more hidden layers in between (in the latter case, the network is called 'deep'). Each layer is composed of one or more neurons, which are working like a biological neuron: Multiple input signals are passed to a neuron. If the aggregated inputs surpass a certain strength, the neuron is activated and transmits a signal to its outputs. For artificial neurons, we assume that the incoming signal (for all but the input layer) is the weighted sum from other neurons' outputs plus a bias. The activation of the artificial neuron happens according to an activation function (e.g. Rectified Linear Unit, ReLU as described in Nair and Hinton, 2010). The different hidden layers are used to learn the true output. This is done by optimizing the weights and the bias for each neuron.

The goal of our neural network is to assign a FAIM image to one of the three label classes "calm", "cloudy" or "dynamic". Hence, the output layer of our NN has three neurons. Each output neuron is representing a label class and is supposed to output



the probability of the respective class given the input features. These three output probabilities are combined in a vector, the prediction vector.

Since considering a sequence of images instead of an individual image often simplifies the discrimination between the different classes, our neural network uses sequences as input. In our case these are sequences of the above-mentioned features and not sequences of images. The sequences of features are derived from several consecutive images that are located symmetrically in time around the original image that is to be classified.

In order to attribute one image to a specific class, we used a Temporal Convolutional Network (TCN, see e.g., Bai et al. 2018) in the TensorFlow Keras implementation of Rémy (2020). TCNs are based on dilated convolutions, so at each neuron of the hidden layer a convolution takes place. In our case, the input is a time series stored in $x$. Each temporal component of $x$ consists of the eight features mentioned before and is calculated from the same image. Thus x is two-dimensional and of size $T \times 8$, with $T$ being the length of the time series. The kernel $k$ is a function (details are given later) with which our input $x$ is convolved. It is two-dimensional with dimension $(2r + 1) \times 8$ : $(2r + 1)$ is its temporal length (i.e., $k$ is defined at $-r, -(r - 1), \dots, -1, 0, 1, \dots, (r - 1), r$, where r can be chosen) and eight is due to the eight features per time step. $d$ is the dilation factor ($d \in \mathbb{N}$).

The dilated convolution is then calculated as follows:

$$x'(t) = (x *_d k)(t) = \sum_{a=-r}^{r} x^T(t - da) \cdot k(a) \qquad (1)$$

The result of the dilated convolution $x'(t)$ is a scalar (in contrast to $x(t - da)$ which is the vector of features at time $t - da$ and $k(a)$ which is a vector of length 8). The dilation factor $d$ leads to the fact, that for the computation of $x'$ not every but every $d^{\text{th}}$ temporal component of $x$ is taken. From the range of the running index $a$, which is going from $-r$ to $r$, it becomes clear that we have here a so-called non-causal convolution, i.e., for the calculation of $x'$ at time $t$ also values of $x$ at time points later than $t$ (i.e, values referring to the future) are included. The values of kernel $k$ represent the weights mentioned at the beginning of this section. So, through the training process, the kernel and therefore the weights as well as the bias are optimized for each hidden layer in order to achieve the true classification.

In TCNs, these dilated convolutions are stacked, which is visualized for two dilations in Figure 2. The dilation factor increases by a factor of 2 with each additional stacked dilated convolution. This makes sure that all information from the input sequence contribute (in a modified way) to the output and allows large input sequences with only a few layers.





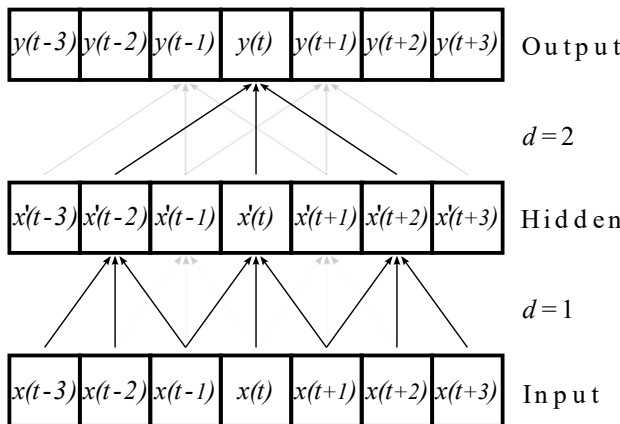

205

**Figure 2: Stacked dilated convolutions applied to a time series $x(t)$. The output is denoted with $y(t)$. The dilated convolutions have the dilation factors $d = 1, 2$ and a kernel of length 3 (according to Bai et al. (2018)). In order to avoid a shortening of the time series in each step, the series are enlarged by zeros at the beginning and the end (also called zero padding).**

We constructed and trained two TCN instances for different sequence lengths. The short sequence includes features of 13 time

210 steps, which corresponds to a time of approximately three minutes. The long sequence includes features of 61 time steps, which

corresponds to a time span of approximately 14 minutes. These two different sequence lengths were used so that one TCN

($TCN_{13}$), on one hand, has a way of reacting well to short-term events. On the other hand, the greater information content of a

long sequence can be used, so that the second TCN ($TCN_{61}$) can better classify unclear episodes. The two sequence lengths

lead to two independent classifications by the respective TCN for the same point in time.

215 We always used a kernel size of 3 for the dilated convolutions. Furthermore, for the input sequence length of 13 the dilation

factors for the $TCN_{13}$ were $d = 1,2$ while for the input sequence length of 61 the dilation factors for that TCN were $d = 1, 2, 4, 8$. Comparing the given sequence length of 13 for the given dilation factors 1, 2, with the sequence length from Figure

2 for the same dilation factors, a difference between the theory and implementation can be noticed. This is a known property

of the given implementation (https://github.com/philipperemy/keras-tcn/issues/207, https://github.com/philipperemy/keras-

tcn/issues/196). The implementation always achieves a maximum sequence length with a dilation factor less than required in

theory. For example, a maximum sequence length of 13 does not require the theoretical dilation factors of $d = 1, 2, 4$, it only

requires $d = 1, 2$.

The number of filters (number of different stacked dilated convolutions applied on every feature sequence) is 16. This results

in approximately 3,000 trainable parameters for $TCN_{13}$ and 6,000 trainable parameters for $TCN_{61}$.


After describing the basic idea of a TCN as introduced in Bai et al. (2018), we also would like to give the most important

information about the implementation of the TCN. For this the TCN used so-called residual blocks as described in Bai et al.

(2018). A residual block consists of two hidden layers (each hidden layer comprises the weighting of the signals using dilated

convolutions, the activation of the neurons and the processed signals) and a skip connection. The skip connection allows to

jump over hidden layers. As activation function we used the rectified linear unit (ReLU) in the residual block (Nair and Hinton,



2010) and due to the classification task the softmax function in the output layer. Softmax ensures amongst others that the individual values of the prediction vector, so the output of the neural network, can be interpreted as a probability. The weights in the residual block are normalized during the training process with weight normalization as introduced in Salimans and Kingma (2016) and for temporal convolution networks suggested in Bai et al. (2018).

One challenge when using neural networks is to avoid overfitting, i.e., the network only memorizes the training data. In order to prevent this, we used a dropout regularization as proposed in Srivastava et. al (2014) with the ratio of 0.3 in the residual block as well as in the layer before the output layer. That means randomly 30% of the inputs of an neurons in each of these layers are switched off during the training of the network. Additionally, Gaussian noise is added to the time series before it is passed to the TCN. The architecture of our TCN is displayed in Figure 3.


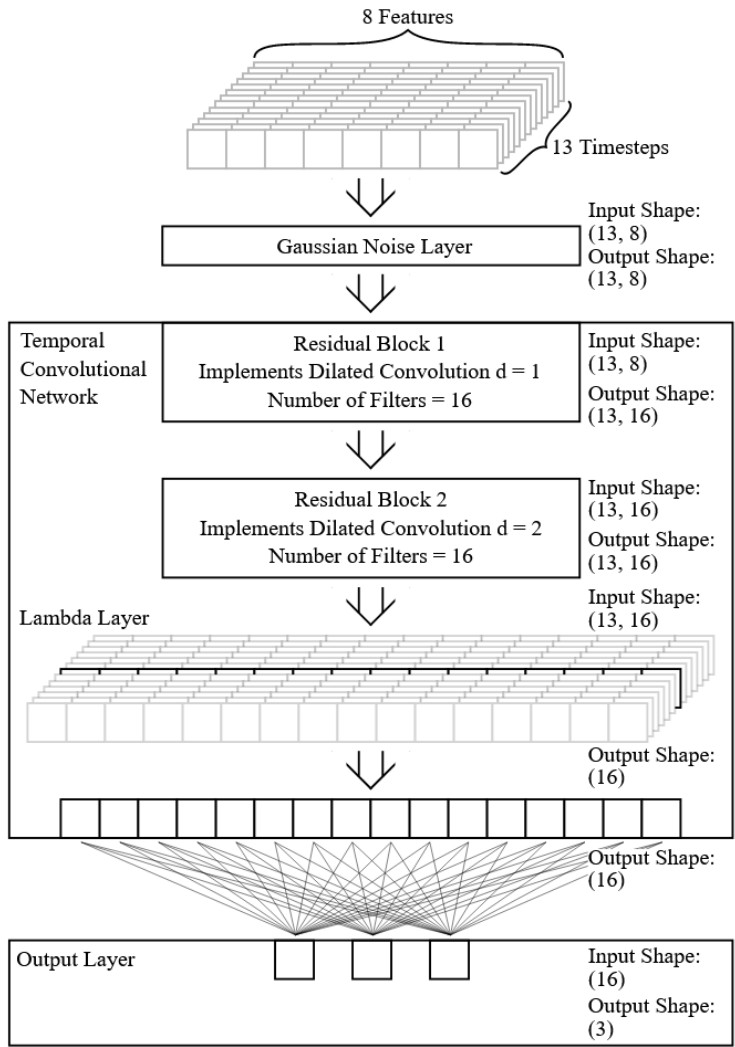

**Figure 3: The input is a sequence of length 13 with 8 features at each time step. It has the dimension 8\*13. The first layer is a Gaussian Noise Layer, which is only active during training and adds a slight normally distributed noise to the input data (standard deviation = 0.01). Afterwards the 8\*13 data points are passed to the TCN and there to the first residual block. This residual block**

**implements a one-dimensional dilated convolution with a dilation factor of d = 1 and kernel length of three. Since we have 16 filters (16 different initialized kernels) this also leads to 16 output features. The number of time steps remains the same. Afterwards, the same is repeated in the second residual block, only with the dilation factor increased to d = 2. The last step in the TCN is picking up the black marked middle element of the 13 time steps since only this element contains information about the complete time sequence. This is done with the help of a so-called lambda layer. Finally, we map the 16 features resulting from the TCN to the 3 output neurons**

**with a fully connected layer and the softmax function as activation function. In this representation, the dropout regularization in the residual blocks as well as in the output layer (with a factor of 0.3) is not shown.**



As mentioned at the beginning of this section, the class prediction of each image is stored in a vector. The length of the vector is equal to the number of classes. Since we have three classes, the vector is three-dimensional. The ground truth classification vector has a single entry equal to one and two entries equal to zero, because each image is manually assigned to a single label class. This kind of classification encoding is called a one-hot classification vector. The prediction vector of our learned classifier also has three entries and every entry gives the predicted probability of the respective label class. We calculate the

mean vector of the two prediction vectors for a sequence length of 13 and 61 and call this the "combined classifier" whose entries also can be interpreted as a probability for the respective label class. To retrieve information about the quality of the classification and to learn, the difference between the classification vector and the prediction vector needs to be measured; this is done by the "categorical cross entropy" metric which is explained in Murphy (2012). This is repeated for all inputs in a batch and the resulting average is called loss. This loss has to be minimized based on the adjustment of the trainable parameters (i.e.,

weights and biases of the different neurons), which can be done using a gradient-based optimizer. Our TCN was trained by the Adam optimizer, which was introduced in Kingma and Ba (2014). A starting learning rate of 0.05 provided the best results. The learning rate was additionally (to the adjustments of the optimizer "Adam") adjusted at the beginning of all 100 training epochs in the following way: After each epoch the learning rate was reduced by a factor of $\exp{(-0.2)}$. After 25 epochs and multiples thereof the learning rate was increased to approximately 70% of the last maxima. This principle of such so-called

cycling learning rate was proposed in Smith (2017) and leads to the fact that only a range around the perfect learning rate has to be found instead of a perfect fitting learning rate.

During the training process we saved the model with the lowest loss on the validation data, also estimated by the categorical cross-entropy metric.

Due to the large amount of data, it is most important that sequences predicted as "dynamic" are actually dynamic episodes. This can be measured by the precision. The precision $P_i$ of a label class $i$ is the quotient of correctly positive predicted time steps of a label class $t_{\mathrm{p}_i}$ and all time steps that are assigned by the classifier to a label class (,i.e., the sum of the correctly positive and the false positive predicted time steps $t_{\mathrm{p}_i} + f_{\mathrm{p}_i}$):

$$P_i = \frac{t_{\mathrm{p}_i}}{t_{\mathrm{p}_i} + f_{\mathrm{p}_i}} \qquad (2)$$

The counterpart to precision is recall $R_i$ (of a label class $i$), which is calculated by dividing the correctly positive predicted time steps of a label class $t_{\mathrm{p}_i}$ by all the time steps manually assigned to a label class (,i.e., the sum of the correctly positive predicted and the false negative predicted time steps $t_{\mathrm{p}_i} + f_{\mathrm{N}_i}$):

$$R_i = \frac{t_{\mathrm{p}_i}}{t_{\mathrm{p}_i} + f_{\mathrm{N}_i}} \qquad (3)$$





Therefore, recall is a measure of how many time steps of a label class are actually recognized by the classifier, whereas precision only evaluates the time steps assigned to a label class by the classifier and thereby determines the proportion of all correctly assigned time steps.

Thresholds can be defined to determine at what value a prediction is assigned to a label class. For example, if a threshold value of zero is set for the output neuron of the label class "dynamic", all time steps will be assigned to the label class "dynamic". In

this case, the recall is at its maximum value of one, whereas the precision is usually at its minimum, since there is normally a large number of false positive time steps. If the threshold value is now increased step by step, recall decreases and precision increases at the same time.

For each threshold value, a value pair of precision and recall can now be formed. Plotting recall versus precision and calculating the area under the curve, we get the so called "average precision" (AP) of a respective label class, which can reach the

maximum value of one (see Figure 4). This is a reliable quality metric for the detection of a label class of a classifier independent to the used thresholds. Calculating the mean value of all average precisions, so for calm, dynamic, and cloudy, gives the mean average precision of a classifier.

**3.5 Analysis of the classification algorithm**

According to the precision-recall curves on the test data set (Figure 4), the combined classifier achieves a mean average

precision of 0.82. Taking a closer look at all average precisions reveals the following result.

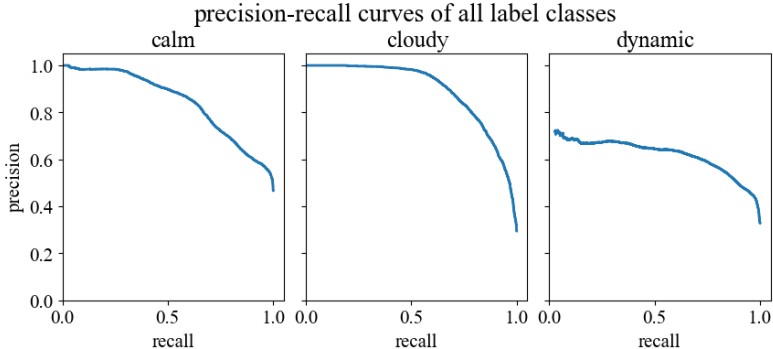

**Figure 4: Precision-recall curves of all three label classes on the test data. Each value pair of precision and recall is based on a threshold value which decides whether a prediction is assigned to a label class or not. These thresholds start at a high level and are decreased constantly until a recall of 1.0 is achieved. This is done for all label classes separately. The area under the precision-recall curve is called average precision (AP) of the label class.**


The average precision for the actual target label class "dynamic" is 0.63. If we consider the same statistical measures on the non-target label classes "calm" and "cloudy", we achieve an average precision of 0.85 for the label class "calm" and an average precision of 0.90 for the label class "cloudy".





For our combined classifier, we used features instead of whole images. In order to determine the importance of individual
features groups for detection, we derived the precision values by omitting one of the three groups of features (basic, texture
and psd features). If we omit the feature groups basic features or texture features, we only detect tiny changes of the average
precision of the label classes "calm" and "cloudy". The decrease of the average precision of the label class "dynamic" is a bit
higher, while omitting the basic feature group instead of the texture feature group. This leads to a slightly lower mean average
precision by omitting the basic feature group compared to the texture feature group.  The greatest influence on all statistical
measures has the psd feature group. Omitting this feature group leads to a decrease of approximately ten percent in every
average precision. Therefore, the mean average precision decreases also by approximately ten percent.

So far, we have reported the general metrics of the neural network. To sum it up, the prediction of the label class "cloudy" has
given the best results, followed by "calm" and with some distance "dynamic".

Since our goal is the identification of dynamic episodes, we adjusted the classification criteria to optimize the precision of the
label class "dynamic". This configuration was evolved on the validation data and tested on the test data in the final step. The
classification criteria are specified as follows:

First, we set thresholds whether predictions are assigned to a label class or not. This is done using Figure 5, which shows the
precision and recall for each label class as a function of the threshold values.

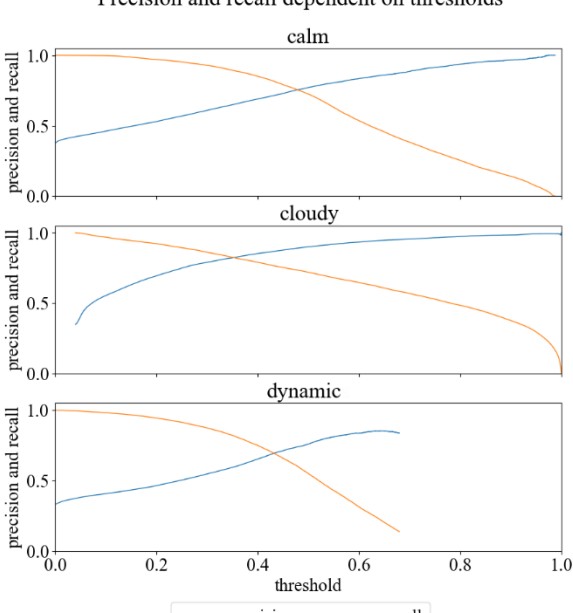


**Figure 5: Precision and recall dependent on thresholds (on validation data) for all three label classes "calm", "cloudy" and "dynamic".**



We have chosen threshold values 0.35 for the label class „clouds" and 0.5 for the label class „calm", since the precision and recall of these two label classes are almost identical for these threshold values on the validation data set. For the label class

„dynamic" we have chosen 0.5 as threshold. In contrast to the other label classes, precision is higher than recall in order to optimize the precision of the label class „dynamic". Due to these individual thresholds, it is possible that the final classifier suggests no or more than one label class for some time steps.

If time steps are not assigned to any label class, we classify them as "unsure". If the classifier suggests more than one label class for a time step, the label class with the highest average precision wins the prediction.

Applying this procedure to the test data leads to the following confusion matrices and statistical measures (Table 1,Table 2 andTable 3).

In confusion matrices, the manual classifications are plotted in horizontal direction and the automatic predictions in vertical direction. Correct predictions are therefore on the main diagonal, whereas incorrect predictions are on the secondary diagonal. This allows to identify label classes that can be well distinguished, but also label classes that are more difficult to distinguish.

|  |  | prediction |  |  |  |  |
|---|---|---|---|---|---|---|
|  |  | calm | cloudy | dynamic | unsure | Sum |
| classification | calm | **6,862** | 540 | 1,982 | 1,081 | 10,465 |
|  | cloudy | 109 | **5,046** | 567 | 646 | 6,368 |
|  | dynamic | 1,402 | 764 | **4,371** | 931 | 7,468 |
|  | sum | 8,373 | 6,350 | 6,920 | 2,658 | 24,301 |

**Table 1: Confusion matrix of the combined classifier, with thresholds of 0.35 for the label class „cloudy", 0.5 for label class „calm" and 0.5 for the label class „dynamic".**

The confusion matrix displayed in Table 1 presents a detailed look on every manual classification and prediction of the combined classifier. As the number of manual classifications differs in each label class, drawing a conclusion on the quality

of the classifier's predictions regarding the confusion of individual label classes is quite difficult. Therefore, we also display a normalized version of the confusion matrix in Table 2. Each row is normalized by the sum of each row, so that the results in Table 2 are independent of the number of manual classifications in each label class.

|  |  | Prediction |  |  |  |
|---|---|---|---|---|---|
|  |  | calm | cloudy | dynamic | Unsure |
| classification | calm | **0.66** | 0.05 | 0.19 | 0.10 |
|  | cloudy | 0.02 | **0.79** | 0.09 | 0.10 |
|  | dynamic | 0.19 | 0.10 | **0.59** | 0.12 |

**Table 2: Normalized confusion matrix of the classifier displayed in Table 1. Each row in the confusion matrix was divided by the**
**sum of the row, which is equal to the sum of classifications in the corresponding label class.**





Before we go into the details of the confusion matrices in Table 1 and Table 2, we shortly focus on the statistical measures which are calculated based on the confusion matrix in Table 1. The statistical measures precision, recall and overall accuracy are presented in Table 3:


|  | calm | cloudy | dynamic |
|---|---|---|---|
| precision | 0.82 | 0.80 | 0.63 |
| recall | 0.66 | 0.79 | 0.59 |
| overall accuracy | 0.67 | | |

**Table 3: Statistical measures "precision", "recall" and "overall accuracy" calculated by the confusion matrix of Table 1.**

As we have seen before, Table 3 also shows that the combined classifier gives the best results for the label class „cloudy",
followed by „calm" and with some distance "dynamic". There is just little difference between precision and recall in both label classes "cloudy" and "dynamic", while the recall of the label class „calm" is much lower than the precision of the label class „calm".

This is initially unexpected, because we adjusted our thresholds in a way that recall and precision on the label classes „calm" and „dynamic" are almost on the same level and for the label class „dynamic". However, this can be explained by the confusion
matrix in Table 1 andTable 2. First, the label classes „calm" and „dynamic" have a high potential of being mixed up: 19% of the episodes classified as „calm" are predicted as „dynamic" and vice versa. Secondly, all label classes occur at similar frequencies in the validation data set. So, if we compare this to the test data set, we can see (in the column "sum") that there are by far more „calm" classifications than „dynamic" classifications. Combining these two aspects, it is on the one hand clear that the precision of the label class is boosted by the large amount of „calm" classifications and therefore higher than the recall
of the label class „calm". On the other hand, a larger number of „calm" classifications leads (due to the high potential of mixed-up predictions between „calm" and „dynamic") to a large number of predictions of „dynamic", which were originally classified as „calm". This decreases the precision of label class „dynamic".

The confusion matrix in Table 1 suggests that there are in total more „dynamic" predictions which are originally classified with the label class „calm" than vice versa. But we have also seen that there are by far more „calm" classifications than
„dynamic" classifications. Due to these aspects the confusion matrix in Table 2 is normalized by the sum of each row, which means by number of classifications of each label class. This confusion matrix shows that the proportion of mixed-up „calm" and „dynamic" timestamps in relation to the number of classifications of each label class is the same in both directions (19 percent of the label classes „calm" or „dynamic"). It also shows that the lower recall of the label class „dynamic" (0.59) compared to the label class „calm" (0.66) is mainly caused by the increased number of „cloudy" predictions in relation to the
total number of classifications, when the timestamps are classified as „dynamic".





It has been shown that the imbalance of the test data set and the mixing-up between „calm" and „dynamic" predictions has a large influence on the statistical measures. Therefore, as a next step, calm predictions which are classified as „dynamic" and „dynamic" predictions which are classified as „calm" will be investigated. All these episodes are categorized with mispredicted or misclassified (Table 4).


| | Classified as "calm" but predicted as „dynamic" | | Classified as "dynamic" but predicted as "calm" | |
|---|---|---|---|---|
| | frequency | relative frequency | frequency | relative frequency |
| mispredicted | 872 | 0.44 | 362 | 0.26 |
| misclassified | 1,110 | 0.56 | 1,040 | 0.74 |
| sum | 1,982 | 1.00 | 1,402 | 1.00 |

**Table 4: Overview of "calm" and "dynamic" episodes categorized in mispredicted and misclassified.**

Table 4 shows that in both cases most of the time steps that are considered as wrongly predicted are actually misclassified and not mispredicted. The relative frequency of mispredicted episodes is higher in the case of classified as „calm" but predicted as „dynamic" than in the opposite case. This can be partly explained by gravity wave structures, which are classified as „dynamic" and often predicted as „calm", which leads to the assumption that our classifier or our features are not able to detect these structures.

In further next step the data classified as "dynamic" is used to find turbulence episodes. To determine the frequency of turbulence episodes in this label class, all sequences of the test data set predicted as „dynamic" were viewed and split into three categories: Turbulence, if rotating structures of nearly cylindrical shape can be detected; potentially turbulence, if rotating structures can be suspected, but not clearly detected; no turbulence, if no structures can be observed that are relatable to rotating cylinders. Examples of these three categories can be seen in Video 1 (see supplements to this article). The intervals of these three categories are discussed in detail within the next section.

| | frequency | relative frequency |
|---|---|---|
| turbulence | 1,825 | 0.26 |
| potentially turbulence | 2,127 | 0.31 |
| no turbulence | 2,714 | 0.39 |

**Table 5: Categorization of all as „dynamic" predicted timestamps into three categories: Turbulence, potentially turbulence and no turbulence.**


Splitting all as "dynamic" predicted timestamps in the described manner delivers three categories of roughly the same size (see Table 5). Slightly less than one third contains turbulence. Another third contains structures that can be related to turbulence and slightly more than one third does not contain any structures that can be related to turbulence.



### 3.6 Discussion of the Classification algorithm

At first glance, the statistical measures of mean average precision of 0.82 and average precisions of 0.90 for the label class „cloudy", 0.85 for the label class „calm" and 0.63 for the label class „dynamic" on the test data set appear satisfying, but not entirely convincing. In this context, a few aspects need to be considered. For instance, the fundamental task is not one of completely unambiguous class assignments (as it is in the case of basic object detections). In our case, the transitions between the label classes, in particular the transitions between "calm" and "dynamic" are fluid. This also means that the manual

classification is always subjective to some degree. Furthermore, there are events that are generally difficult to classify manually, such as very short cloud fields moving rapidly through the field of view, or short „calm" (non-cloudy) episodes between cloud fields. To illustrate these aspects with an example, a video is digitally attached to the submission (Video 1). It shows the complete video footage of one night out of the test data set and is provided with the manual classification as well as with the raw and final automated prediction of the classifier (i.e., the manual classification is represented as a one-hot vector

and the raw prediction is represented as a one-hot vector). In this video, some turbulent vortices are visible (19:30, 19:55, 20:10, 20:50). These time periods are all correctly predicted as "dynamic". Furthermore, the combined classifier detects even the smallest cloud veils, such as from 19:44–19:52. These are undoubtedly detectable, but only faintly and briefly, so they were not noticed in the manual classification. In the statistics, this is counted as a false classification, even though the labelling was incorrect.

The largest impact on the statistical measures is the confusion between „calm" and „dynamic" episodes (as shown in Table 1 and Table 4). Video 1 in the supplements shows also confused „calm" and „dynamic" episodes, especially episodes which are classified as „dynamic" but predicted as „calm" (22:06–22:14 and 22:23–22:45). Taking a closer look at these episodes reveals the difficulty of drawing boundaries between the label classes „calm" and „dynamic". Nevertheless, episode 22:06–22:14 and 22:33–22:45 can be considered misclassified, whereas the episode 22:23–22:33 is more likely to be considered as mispredicted.

Although only two of three episodes can rather be considered as misclassified, all three episodes are counted as misclassification and formally impair the statistical measures of the classifier. So, in these cases it is mainly the manual classification that is wrong and not the prediction of the classifier. The prediction of the classifier provides also further advantages: On the one hand, the prediction of the classifier happens without significant time effort, whereas the manual classification of future data would take an extreme amount of time. On the other hand, it is not affected by human effects such

as lack of concentration and subjectivity. This leads to the result in Table 4, which implies that the majority of confused calm and "dynamic" timestamps are caused by the manual classification due to misclassifications instead of mispredictions of the classifier. This leads to the fact that the classifier is better suited to distinguish between "calm" and "dynamic" episodes than the manual classification. These confusions due to misclassification have the largest negative impact on precision and recall of the label class "dynamic" (listed in Table 3 and calculated according to Table 1) and are caused by errors in the manual

classification (Table 4), not by mispredictions of the classifier.





Assuming that the validation and training dataset containing the same proportion of "calm" and "dynamic" episodes that have been mixed up during the manual classification, it is worth saying that the training and validation data do not have to be perfectly classified (manually) in order to train a well performing classifier.

The classification into three different label classes "calm", "dynamic" and "cloudy" is a natural approach according to the video 440 material. This does not automatically mean, that this is a helpful search space restriction for the automated search for turbulence.

Looking at the relative frequencies of turbulence and potential turbulence (Table 5) reveals that about two thirds of all data belonging to the label class "dynamic" can be of interest for turbulence analysis.

In the video shown in the supplement, the intervals 19:28–19:30, 20:00–20:08, 20:22–20:25, 21:16–21:29 are annotated with 445 potential turbulence and the intervals 19:30–19:43, 19:52–20:01, 20:08–20:22, 20:50–20:58 are episodes in which turbulence can be observed. In order to complete the annotations of the video, the sequences that have been annotated with no turbulence are the following: 20:25–20:35, 20:48–20:50, 20:58–21:02, 21:05–21:17, 21:30–22:05, 22:12–22:23. It remains to be said that the classification into the three label classes "calm", "dynamic" and "cloudy" can be regarded as quite reasonable since in two thirds of all "dynamic" timestamps either turbulence or events resembling to turbulence occur.

In future work, the resulting downsized dataset (all data that are predicted as "dynamic") can now be used to train neural networks using image data to directly detect and potentially measure turbulence. This was not possible before due to several issues:

Firstly, the computational cost of processing image sequences in a neural network is higher than processing a time series of manually determined features of the images: a time series of images requires processing 10,000 data points per time step (if 455 the image has a resolution of 100*100 pixels), whereas a time series of our features only requires processing 8 data points per time step. Using time series (with 13 resp. 61 time steps), instead of a single time step increases computational costs additionally. This will become crucial when applying this method to larger amounts of data.

Secondly, training with single images instead of image sequences to reduce computational cost is not ideal. Image sequences, in comparison to single images, are containing essential information for the differentiation of the respective label classes. For 460 example, cloud veils in a single image often cannot be distinguished from the label class "dynamic", or turbulence vortices that appear similar to rotating cylinders can only be clearly identified by the information of the image sequence.

This reduced dataset contains only episodes that show „dynamics" in the UMLT, of which approximately two thirds are potentially related to turbulence. Thus, the dataset is more balanced with respect to turbulence, which simplifies training for direct search and measurement of turbulence. Future work will also not waste computational time on "calm" and "cloudy" 465 episodes (where observable turbulence is not expected), making training with the image sequences more efficient.





## 4 Analysis and discussion of turbulence

Checking the "dynamic" episodes from the TCN model by hand we identified 19 episodes of turbulence. 13 of them exhibited

such a good quality that we were able to read out the length scale and the root-mean-square velocity of turbulence and calculate

the energy dissipation rate $\epsilon$, according to the method applied by Hecht et al. (2021) and Sedlak et al. (2021). The resulting

values are displayed in Table 6. The uncertainty $\delta\epsilon$ is calculated by applying the rules of error propagation to equation (1).

Similar to Sedlak et al. (2021), a general read-out error of $\pm 3$ pixels is used. This leads to an uncertainty of the length scale $\delta L$

of $\pm 69$ m and (since velocities are determined over a set of ten images) an uncertainty of the velocity $\delta U$ of $\pm 2.5$ m s$^{-1}$.


| Date | $\epsilon$ [W kg$^{-1}$] | $\delta\epsilon$ [W kg$^{-1}$] |
|---|---|---|
| 16 June 2019 | 0.55 | 0.50 |
| 16 June 2019 | 0.46 | 0.44 |
| 16 June 2019 | 2.24 | 1.43 |
| 6 July 2019 | 0.06 | 0.11 |
| 18 July 2019 | 0.19 | 0.24 |
| 18 July 2019 | 0.03 | 0.09 |
| 3 September 2019 | 2.43 | 1.55 |
| 13 October 2019 | 1.75 | 1.09 |
| 13 November 2019 | 3.18 | 1.39 |
| 12 December 2019 | 2.65 | 1.25 |
| 18 December 2019 | 0.78 | 0.63 |
| 29 December 2019 | 0.19 | 0.24 |
| 19 February 2020 | 0.04 | 0.09 |

**Table 6: Values of energy dissipation rate $\epsilon$ and the corresponding uncertainty $\delta\epsilon$ of all turbulence events of sufficient quality found in the airglow image data between 11 June 2019 and 25 February 2020 acquired at Oberpfaffenhofen, Germany.**

The values of $\epsilon$ range from 0.03 to 3.18 W kg$^{-1}$ with a median value of 0.55 W kg$^{-1}$ and a standard deviation of 1.16 W kg$^{-1}$. In

Sedlak et al. (2021), for comparison, $\epsilon$ ranges from 0.08 to 9.03 W kg$^{-1}$ with a median value of 1.45 W kg$^{-1}$. In both studies, the

values cover three orders of magnitude, however this is also reflected in the publications of other authors. Hecht et al. (2021)

found an energy dissipation rate of 0.97 W kg$^{-1}$ with this approach; Chau et al. (2020) present an energy dissipation rate of

1.125 W kg$^{-1}$ and claim that this would be a rather high value. Hocking (1999) finds a maximum order of magnitude of 0.1 W kg$^{-1}$.





Although the identification of turbulent episodes is done automatically via the TCN approach presented in section 3 the measurement of $L$ and $U$ is still done manually. This implies an inherent read-out uncertainty due to the blurry structures and a remaining possibility of misinterpretation, which we intended to minimize by multiple-eye inspection of the episodes. It is interesting to note that no events with very large energy dissipation rates in the range $4$–$9\,\mathrm{W\,kg^{-1}}$ as in Sedlak et al. (2021) were found in this data set, whereas the lower limit of $\epsilon$ is quite similar. It is still an open question whether the larger values are the

result of direct energy dissipation or if the respective large eddies are about to further decompose to smaller structures beyond the sensitivity of our instrument, which then mark the actual end of the energy cascade.

The values in Table 6 are in good agreement with literature values, however the data basis is still very small. Future measurements of airglow imagers will have to be analyzed with the method applied here in order to establish resilient statistical conclusions.

**5 Summary and outlook**

We have investigated the application of practical and easy-to-use algorithms based on Neural Networks (NN) to facilitate the detection of episodes showing turbulent motions in OH* airglow image data. This is done by setting up two variants of a TCN (Temporal Convolutional Neural Network) to automatically pre-sort the images into images exhibiting strong airglow „dynamics" (where turbulence can likely be found), and images exhibiting calm airglow „dynamics" or being disturbed by

„clouds" (which can be excluded from further turbulence analyses). The image data used in this work has been acquired by the high-resolution camera system FAIM 3 at Oberpfaffenhofen, Germany between 11 June 2019 and 25 February 2020.

The TCN-based classification algorithm (based on the time series of features derived from the temporal image sequences) achieves a mean average precision of 0.82. We demonstrated with a video example from the test data set that the algorithm works much better than the statistical values suggest. All in all, 13 episodes exhibit a sufficiently high quality to derive the

energy dissipation rate $\epsilon$. Values range from 0.03 to $3.18\,\mathrm{W\,kg^{-1}}$ and are in good agreement with previous work. The data analyzed here confirms the importance of considering dynamically driven energy deposition by breaking gravity waves when studying the energy budget of the atmosphere.

We have shown that a NN-based algorithm can support the identification of turbulent episodes in airglow imager data. This marks an important step to expand the method of extracting turbulence parameters from airglow images from local case studies

to investigations of global extent. Utilizing Neural Networks is a promising way of dealing with 'big data'. With ongoing airglow measurements, it will be possible to also investigate effects like seasonal or latitudinal variations of the energy dissipation rate with diminishing uncertainties.





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

**Figures**

**Authors contributions**

The conceptualization of the project, the funding acquisition, and the administration and supervision were done by MB and
SW. The operability of the instrument as well as image data preparation was assured by RS. The TCN algorithm was adjusted, tested and applied to the image data by AW. The image features were calculated by RS with the calculation of the psd features basing on results of the 2d-FFT algorithm as presented by PH in Hannawald et al. (2019). The analysis of turbulence was done by RS and AW. The analysis of the algorithmic performance was done by AW. The interpretation of the results benefited from fruitful discussions between AW, PH, SW, MB and RS. The original draft of the manuscript was written by RS and AW.
Careful review of the draft was performed by all co-authors.