# Peer review of "Analysis of 2D airglow imager data with respect to dynamics using machine learning"

_Atmospheric Measurement Techniques, 2023_

## Referee Comment (RC1)

Atmospheric Measurement Techniques (AMT)

Journal: AMT
Title: Analysis of 2D airglow imager data with respect to dynamics using machine
        learning
Author(s): René Sedlak, Andreas Welscher, Patrick Hannawald, Sabine Wüst, Rainer
        Lienhart, and Michael Bittner
MS No.: amt-2023-25
MS Type: Research Article

General Comments

1. The authors describe the development and results of applying a neural network based classification method to sequences of OH* nightglow images obtained from a very high resolution InGaAs-based camera.  The goal of the work was to find an efficient method of identifying images that captured turbulence or had the potential to show turbulence from which the energy dissipation rate in the turbulence could be calculated.  In order to achieve this goal, the classification scheme would segregate images into one of three categories namely "cloudy", "dynamic" and "calm".  Only the dynamic images carried the information sought.  The variation in images that display turbulence in terms of the scale size and persistence makes the task of identifying dynamic images very challenging.

2. As explained by the authors, attempting to develop a classification method by applying an artificial intelligence approach directly to the images would be computationally very demanding.  An alternative approach based on eight features (two basic; three textual, and three based on power spectral density calculations) representing each image was employed as the input to a neural network (NN).  The features chosen together with the characteristics they represent are clearly described.  Sequences of these image features, rather than features of individual images, are the input to the neural network because this helps the classification process – just as it does in the case of manual classification.  Two sets of sequences are used – a 13 image sequence and a 61 image sequence (both chosen symmetrically around the time of the image to be classified) – to provide sensitivity to both short-term and longer term events.  Thus two neural network (Temporal Convolution Network (TCN)) instances were constructed.  The output of the two neural networks was combined to calculate a probability for each of the label classes (cloudy, dynamic or calm).

3. The manuscript is well structured and clearly written.  It provides a great deal of detail on the classification algorithm employed and the architecture of the TCNs as well as the training procedure and evaluation of the results.  It is to the credit of the authors that the high level of technical detail and the reasons for the decisions are clearly explained.

4. Cameras of the type used in this study are being employed more and more frequently in recent years for atmospheric studies.  They produce enormous quantities of data, which mush be archived and then analysed thereby making the work described here very interesting and a valuable contribution to this field of study.  The text includes an appropriate set of references.  The work is suitable for publication in AMT, provided that the following points are addressed.

Specific Comments
1. Explain the rationale behind the decision to use 70% of the available dataset for training; 20% for validation and 10% for testing.

2. Explain the reason for 100 epochs (line 157) (see also lines 267-268). What determines the number? For example, could it be 50 or 200?

3. One can readily appreciate the difficulty of manual classification as discussed in line 409 and following. However, the number of time steps in the test dataset that were deemed to be misclassified (1110 and 1040) in Table 4 is a cause of concern. If I understand this correctly, these were manual classifications. The manuscript describes the negative impact of these manual misclassifications on the statistical measures of the neural network classifier (lines 425-427; lines 430-435 and lines 503-504). Why do the authors not repeat the calculation of the statistical measures using the "correct" classification to establish the "true" value of the statistical measures?

4. Lines 430-435 state that the NN-classifier is superior to manual classification at distinguishing between "calm" and "dynamic" episodes, which is indeed good news for the method, but leaves the reader wondering if the statistical measures have a great deal of validity.

5. A second even more disturbing issue arises with the large number of incorrect manual classifications. Since the test data is only 10% of the total (70% training; 20% validation in line 152), perhaps a large proportion of the training and validation were manually classified incorrectly to start with, thereby having a negative effect on both the training and the validation.

Minor technical corrections and suggestions
Title; Possible alternative title: "Application of machine learning for the detection of turbulence in 2D airglow imager data"
Line 26; Omit b in "2017b".
Line 29/30; replace "OH* measurements are also possible from satellite (see table 1 of Wüst et al. (2023) for limb instruments). They can be made in limb or nadir. " by "OH* measurements are also possible from satellite where they can be made in limb or nadir viewing geometry (see table 1 of Wüst et al. (2023) for limb instruments)."
Line 38; replace "Former studies " by "Previous studies".
Line 42; suggest "propagate" in place of "exist".
Line 54; consider "reliable" in place of "resilient".
Line 69; replace "disbalance" by "imbalance".
Line 71; suggest "very difficult" in place of "hardly possible".
Line 73; replace "data basis" by "database".
Line 81; suggest "The system has been described already … " in place of "The system is already described…".
Line 83; insert a space between "640" and "pixels".
Line 88; insert a space after "175" and "13.9". Replace "pixels$^{-1}$" by "pixel$^{-1}$".
Line 89; insert a space before "km".
Line 90; replace "data basis" by "database". Replace "June, 11th 2019 and February, 25th 2020." by "June 11$^{th}$, 2019 and February 25$^{th}$, 2020.
Line 91; replace "measurement have been performed in 258 nights" by "measurements have been performed on 258 nights".

Line 92; replace "cloud coverage and do not allow" by "cloud cover and prevent".

Line 93; insert "of" after "all".

Line 94; replace "one night has more than one clear episode as soon as it is interrupted by cloudy episodes." by "a single night may have several clear episodes interspersed with cloudy episodes.".

Line 97; recommend "the average image of each episode (a pixel-wise mean of all images in that episode)" in place of "the average image, a pixel-wise mean of all images, of each episode"

Line 107; insert "(compared with structures in "dynamic" class)" after "structures".

Line 117; replace "distinguishing" by "to distinguish".

Line 122; Replace "Mean and standard deviation of the label class "dynamic" are to be expected between "calm" and "dynamic"." by "Both mean and standard deviation of the label class "dynamic" are expected to have values intermediate between those of the "calm" and "cloudy" class."

Line 125: replace "and a way" by "and is a method".

Line 133; insert a space before "pixels".

Line 135; replace "power spectral density (PSD) in dependence of the horizontal wave number $k$ is derived." by "power spectral density (PSD) as a function of horizontal wave number $k$ is derived."

Line 140; consider "takes into account the fact that clouds tend to cause stronger fluctuations over time than during clear sky episodes." in place of "takes the fact into account that, for example, clouds are causing stronger fluctuations over time than during clear sky episodes."

Line 157; Explain the reason for 100 epochs. See also lines 267-268. What determines the number?

Line 162; consider "inadvertent" instead of "unnoticed".

Line 168; consider "exceed" instead of "surpass".

Line 196; omit "also".

Line 237; omit "an" before "neurons".

Line 291; omit the space in the word "precision".

Line 328 and throughout the manuscript; adopt a consistent approach to the use of double quotation marks around the various classes, e.g., "clouds", not „clouds".

Line 332; replace "no or more than one" by no label class or more than one".

Line 335; insert space in "Table 1,Table 2 andTable 3" as follows: "Table 1, Table 2 and Table 3".

Line 337; this sentence appears to be incorrect based on the layout of Table 1 and Table 2 and is a source of considerable confusion for the reader until she/he realizes that it is incorrect. It should state "the manual classifications are plotted in the vertical direction and the automatic predictions in the horizontal direction". To make both Tables completely clear, the left hand column should be labelled "Manual classification", and the top row should be "Classifier prediction".

Line 365; insert a space before "Table 2".

Line 368; omit the word "by".

Line 382; ""calm" predictions" rather than "calm predictions"; i.e., put quotation marks around "calm".

Line 393; insert "a" before "further".

Table 5; how do the values in column 2 (total 6666) relate to the values in Table 1?

Line 428; omit "On the one hand,".

Line 429 replace "On the other hand" by "In addition".

Line 456; replace "(with 13 resp. 61 time steps)" by "(with 13 and 61 steps respectively)".

Line 483/484; the superscript "-1" in "kg$^{-1}$" has become separated onto two lines. Please correct this.

Line 492; replace "data basis" by "database".

Line 493; consider "reliable" in place of "resilient".

Line 509/510; omit "from local case studies to investigations of global extent". Despite the reference to "global information" in line 54, this manuscript does not address this issue in a substantive way.

Line 541; "Horak" should begin on a new line.

Line 552; "Marsh" should begin on a new line.

Line 555; "Murphy" should begin on a new line.

Line 607; replace "basing" by "based".

---

## Author Comment (AC1)

**Authors' Response to Referee Comment #1**

*Sedlak et al., 2023: Analysis of 2D airglow imager data with respect to dynamics using machine learning*

We would like to thank Anonymous Referee #1 for his valuable comments.

**Referring to the specific comments:**

1. Explain the rationale behind the decision to use 70% of the available dataset for training; 20% for validation and 10% for testing.

   These are typical sizes, the exact one is heuristic. But in general, the training dataset should be by far the largest, because the weights are actually adjusted based on the samples of the training dataset, and the validation dataset is larger than the test dataset, because it is used to evaluate all possible adjustments during and after training. Thus, it is used quite often and should contain a large variety of examples, and the test data set is used only as a final check to prevent overfitting on the validation data and thus confirm that the performance on the validation data set can be generalized.

2. Explain the reason for 100 epochs (line 157) (see also lines 267-268). What determines the number? For example, could it be 50 or 200?

   During training, the performance on the validation dataset is monitored, and for both models, most of the improvement happens within the first 50 epochs, and there is barely any improvement after that. So, we could have chosen fewer epochs, but it doesn't matter because the final model is chosen on the best configuration (model of the epoch with the best performance on the validation dataset) during training. More than 100 epochs wouldn't be useful because, as mentioned before, there is hardly any improvement after the first 50 epochs.

3. One can readily appreciate the difficulty of manual classification as discussed in line 409 and following. However, the number of time steps in the test dataset that were deemed to be misclassified (1110 and 1040) in Table 4 is a cause of concern. If I understand this correctly, these were manual classifications. The manuscript describes the negative impact of these manual misclassifications on the statistical measures of the neural network classifier (lines 425-427; lines 430-435 and lines 503-504). Why do the authors not repeat the calculation of the statistical measures using the "correct" classification to establish the "true" value of the statistical measures?

   In Table 4, we only looked at the confused calm and dynamic sequences, but in order to correct the statistical measures, we would have to analyze:
   - all the other dynamic sequences that are not predicted to be dynamic
   - all the other sequences predicted to be dynamic but not classified as dynamic
   - at least all correct calm and dynamic predictions, if they are really correct, since both the manual classifications and the model predictions tend to confuse calm and dynamic episodes.
   If we had adjusted the statistical measures without doing this, we would have biased the statistical measures in a particular direction.

Analyzing this by hand is again very time consuming (much more time consuming because it affects a large portion of the test data) and does not improve the model, and we wanted to invest that time in improving the model.

4. Lines 430-435 state that the NN-classifier is superior to manual classification at distinguishing between "calm" and "dynamic" episodes, which is indeed good news for the method, but leaves the reader wondering if the statistical measures have a great deal of validity.

   That's true, and we tried to emphasize that in our discussion. But the statistical measures still give an indication of the strengths and weaknesses of the model.

5. A second even more disturbing issue arises with the large number of incorrect manual classifications. Since the test data is only 10% of the total (70% training; 20% validation in line 152), perhaps a large proportion of the training and validation were manually classified incorrectly to start with, thereby having a negative effect on both the training and the validation.

   That's also something that we tried to emphasize in our discussion, because it's actually a really good thing. Because it shows that you can train a model that works very well even if you don't have perfectly classified datasets.

**Referring to the minor corrections:**

Title: The suggested alternative title would be another good choice. We decided to leave out title unchanged.

Line 26: 2017b changed to 2017

Line 29/30: Changed.

Line 38: Changed.

Line 42: ‚exist' replaced by ‚propagate'

Line 54: Changed.

Line 69: Changed.

Line 71: Changed.

Line 73: Changed.

Line 81: changed to ‚has already been described in'

Line 83: There is a half-sized space between 640 and pixels which we use as a separator between numbers and units throughout the article. No changes made.

Line 88: Usage of half-sized space: see comment to Line 83. We changed ‚pixels' to ‚pixel'.

Line 89: See Line 83.

Line 90: Corrected. We replaced the date representation as suggested.

Line 91: Following the suggestion of Referee #2 we changed the wording to „during".

Line 92: Changed.

Line 93: Inserted ,of'.

Line 94: Changed.

Line 97: Changed.

Line 107: Changed.

Line 117: Changed.

Line 122: Changed.

Line 125: Changed.

Line 133: See Line 83.

Line 135: Changed.

Line 140: Changed.

Line 157. See answer to Specific Comment #2 above.

Line 162: Changed.

Line 168: Changed.

Line 196: Omitted.

Line 237: Omitted.

Line 291: There is no space in our original file hence this must have been an artifact due to pdf export.

Line 328: We replaced all lower " by upper ones.

Line 332: Changed.

Line 335: Corrected.

Line 337: We corrected this as suggested.

Line 365: Corrected.

Line 368: Omitted.

Line 382: Changed.

Line 393: Corrected.

Table 5: Thank you for finding this inconsitency! We updated the values in Table 6 (former Table 5).

Line 428: Changed.

Line 429: Changed.

Line 456: Changed.

Line 483f: Corrected.

Line 492: Changed.

Line 493: Changed.

Line 509f: Changed.

Line 541: Changed.

Line 552: Changed.

Line 555: Changed.

Line 607: Changed.

---

## Author Comment (AC2)

**Authors' Response to Referee Comment #2**

*Sedlak et al., 2023: Analysis of 2D airglow imager data with respect to dynamics using machine learning*

We would like to thank Anonymous Referee #2 for his valuable comments.

**Referring to the general comments:**

- Out of curiosity, is there a specific reason for pointing the imager toward azimuth 204 degrees?

The azimuthal direction results from aligning the camera system with the window in our laboratory.

- l. 95-98: hasn't fladfielding already removed reflection of the lens for example?

It has for the major part but in practice, there are light remnants of a pattern left which we get completely rid of by subtracting the average image.

- Do you know if the star field has any effects on texture-based parameters like homogeneity or uniformity? Or on the FFT analysis?

Up to now we have not evaluated the effects of the star field on the texture-based parameters, but this might be interesting to investigate. We keep this in mind for our next analyses. We did investigate the influence of the star field on the PSD using quite comparable data captured at Otlica, Slovenia – we did the analysis both with and without removing the stars from the image. It turned out that there is hardly no difference in the course of the integrated PSD except a small offset. The temporal courses of IPSD with and without star removal show a correlation of 0.99, which led us to the assumption that the effect of the star field is negligable for our purpose.

- A table summarizing the 8 features would be helpful to remember what they are, and in which way they characterize the images.

We agree and included such an overview as Table 1.

- l. 268: How did you select the value -0.2? Does it come from another study or is it empirical?

It was empirical, but we paid attention to the range of learning that is covered. And that shouldn't be too high, because otherwise the training becomes ineffective. However, we wanted to have scheduling because it gives better results than without scheduling.

- l. 309-316: The authors say that only the psd feature group shows some significant effect on the precision. Would it be worthy to remove the basic feature group, for example (faster calculations with still good results)?

Would be an option

- Tables 1 and 2 could be combined in only one table XXXX (0.YY)

We appreciate the suggestion but we decided to display them separately for better readability.

- You realized that a lot of the mispredicted "calm" or "dynamic" images are in fact misclassified. How could you improve the classification?

First of all, we were surprised that there were so many misclassifications in the manual classification by hand, but it's maybe reasonable if you consider the following two aspects. First, distinguishing between " calm " and " dynamic " is not like recognizing a clear object. The transitions are smooth, and you are probably (unintentionally) biased by the overall night. For example, on a very calm night you are more likely to consider low dynamic activity as dynamic than on very dynamic nights. On the other hand, it is very hard to stay focused all the time if you are watching these videos for many hours. It is therefore very good news that a large proportion of the mispredicted sequences are in fact misclassified, despite the fact that the model is trained with probably many misclassified images in the training dataset. The aforementioned human bias with respect to single nights is also something that does not play a role in the predictions of the trained model. Therefore, you can improve the classifications by using the model instead of classifying the images by hand.

- The mean average precision is 0.82, but only 0.63 for "dynamic" images (which is what you are looking for). It doesn't seem that good!

Yes, it doesn't look good at first, but we tried to discuss it in detail.

1. There are many misclassified images that affect the result.
2. Distinguishing between calm and dynamic (which is the reason for the low average accuracy) is not like distinguishing between clearly defined objects. Very often the prediction is formally classified as wrong, but one could discuss who is wrong, the manual classification or the model prediction.
3. Nevertheless, the model still confuses a lot of sequences, so more research is needed to find additional features or other ways to minimize this confusion.

- The authors say that only using 8 features in the neural network is faster than the full images, but calculating these features requires already a lot of calculations (like for the FFT).

Indeed, these features require a lot of computation, but you only need to do this computation once because you can store features (the results of these computations). The difference with training a neural network is that the computations for training are needed in every epoch, and in addition they have to be done for whole sequences and not for single frames. These two aspects significantly increase the computational cost compared to our approach.

- Would it be possible to apply this method, or a similar method, on raw data? OF course, it would be a problem for the psd group! So, maybe not possible.

As the psd-based features turned out to be of great importance for the algorithm, we agree that it seems unlikely for this to work. However, we will give this a try in the next analyses. Since the algorithm turned out to perform this will, maybe it might surprise us at analysing images that are unrecognizable for the human eye.

- Only 13 dynamic features in 8 months (~1.5 per month). How does it compare with previous studies from the same authors?

Comparing this to our study of data captured at Otlica, Slovenia (Sedlak et al., 2021), where we could analyse 25 turbulence events in 19 months (~1.3 per month), this rate does not appear to diverse. The measurement setups were also quite comparable as concerns field-of-view and spatio-temporal resolution. With ongoing measurements it will be interesting doing longer-term analyses at different locations and to investigate, whether there are systematic differences in the occurence / intensity of turbulence related to local peculiarities (e.g. the strong bora winds near Otlica). The AI algorithm presented here provides a valuable instrument to approach this research.

**Referring to the minor edits:**

Line 26: Changed to emission.

Line 31: Corrected.

Line 43: Comma inserted.

Line 52: Comma removed.

Line 59: Corrected.

Line 62: Corrected.

Line 69: Corrected.

Line 72: Unfortunately we cannot comprehend what shall be suggested here. No changes made.

Line 80. Parenthesis omitted.

Line 90: Changed.

Line 91: Changed.

Line 94: Already deleted following the suggestions of Referee #1.

Line 92 (we believe that you mean line 97): We indeed mean the window of the building which can cause reflections.

Line 122: Corrected.

Line 129: Changed.

Line 140: Already omitted while following the suggestions of Referee #1.

Line 150: There is no space in our original DOCX file. This seems to be an artifact of the PDF export.

Line 155: Corrected.

Line 237: Already corrected to „of neurons" while following the suggestions of Referee #1.

Line 315: Changed.

Line 321: Changed.

Line 336: Corrected.

Line 365: Corrected.

Line 393: Already changed to „In a further step" while following the suggestions of Referee #1.

Line 419: We changed "even though the labelling was incorrect" to "as the labelling was incorrect".

Line 430: Corrected.

Line 440: Corrected.

Line 492: Changed.

Line 499: Changed.

Line 504: Removed.